# Cancer Immunotherapy with Immune Checkpoint Inhibitors-Biomarkers of Response and Toxicity; Current Limitations and Future Promise

**DOI:** 10.3390/diagnostics12010124

**Published:** 2022-01-06

**Authors:** Brian Healey Bird, Ken Nally, Karine Ronan, Gerard Clarke, Sylvie Amu, Ana S. Almeida, Richard Flavin, Stephen Finn

**Affiliations:** 1School of Medicine, University College Cork, T12 K8AF Cork, Ireland; 2Bon Secours Hospital, T12 K8AF Cork, Ireland; 3School of Biochemistry and Cell Biology, University College Cork, T12 K8AF Cork, Ireland; k.nally@ucc.ie; 4APC Microbiome Ireland, University College Cork, T12 K8AF Cork, Ireland; g.clarke@ucc.ie (G.C.); a.almeida@ucc.ie (A.S.A.); 5Department of Oncology, St. Vincent’s University Hospital, D04 T6F4 Dublin, Ireland; karine.ronan@ucdconnect.ie; 6Department of Psychiatry, University College Cork, T12 K8AF Cork, Ireland; 7Cancer Research at UCC, University College Cork, T12 K8AF Cork, Ireland; sylvie.amu@ucc.ie; 8Department of Histopathology, Trinity College Dublin, D08 NHY1 Dublin, Ireland; rflavin@stjames.ie (R.F.); Stephen.Finn@tcd.ie (S.F.); 9St. James’s Hospital Dublin, D08 NHY1 Dublin, Ireland

**Keywords:** cancer, immunotherapy, biomarker, microenvironment, microbiome, flow cytometry, cytokine

## Abstract

Immune checkpoint inhibitors are monoclonal antibodies that are used to treat over one in three cancer patients. While they have changed the natural history of disease, prolonging life and preserving quality of life, they are highly active in less than 40% of patients, even in the most responsive malignancies such as melanoma, and cause significant autoimmune side effects. Licenced biomarkers include tumour Programmed Death Ligand 1 expression by immunohistochemistry, microsatellite instability, and tumour mutational burden, none of which are particularly sensitive or specific. Emerging tumour and immune tissue biomarkers such as novel immunohistochemistry scores, tumour, stromal and immune cell gene expression profiling, and liquid biomarkers such as systemic inflammatory markers, kynurenine/tryptophan ratio, circulating immune cells, cytokines and DNA are discussed in this review. We also examine the influence of the faecal microbiome on treatment outcome and its use as a biomarker of response and toxicity.

## 1. Introduction

Immune checkpoint inhibitors (ICI) have revolutionised the treatment of cancer. These drugs are monoclonal antibodies that inhibit negative feedback signals in tumour specific T cells. However, most patients do not benefit from these expensive drugs, and a substantial minority experience immune-related autoimmune side effects (irAE) as the unleashed immune system targets other organs. Patients who experience irAE are more likely to have a clinical benefit, but the relationship is not absolute [1]. There are three approved tissue biomarkers in clinical practice, but they lack predictive value of benefit, have low negative predictive value, and do not predict toxicity [2,3]. This review will discuss existing and future biomarkers of response, resistance and toxicity.

There are two classes of licenced ICI. Ipilimumab and Tremelimumab both target the Cytotoxic T-Lymphocyte Associated 4 (CTLA-4) immune checkpoint molecule on T cells activated after encountering professional antigen-presenting cells (APCs) bearing tumour antigens in lymph nodes, leading to clonal expansion of T cells [4]. The second type of ICI targets either Programmed Death Receptor-1 (PD-1) on mature cytotoxic CD8^+^ T cells or its ligands PD-L1 and PD-L2 expressed by cancer cells, stromal cells, and immune cells in the tumour microenvironment. These drugs include the PD-1 antagonists Cemiplimab, Dostarlimab, Pembrolizumab, and Nivolumab, as well as the PD-L1 antagonists Atezolizumab, Avelumab, and Durvalumab. PD-L1 and PD-L2 on binding to PD-1 on CD8^+^ T cells can trigger apoptosis or T cell exhaustion. Inhibition of PD-1 signalling by ICI allows the tumour-specific T cell to become fully activated and attack the tumour cell. This is a simplified reduction of a complicated process. CTLA-4 is also expressed by activated and regulatory T cells, PD-L1 is found on APCs and PD-1 can be found on CD4^+^ T helper cells and B lymphocytes [5].

Each tumour microenvironment (TME) is uniquely heterogeneous both in terms of the composition and spatial distribution of diverse cell populations-which include tumour, stromal and immune cells-and the tumour extracellular matrix (Figure 1). Tumour stromal cells include fibroblasts and mesenchymal stromal cells and provide nutrition to cancer cells [6]. Stromal cells may inhibit tumour specific T cells from infiltrating the tumour or parts of the tumour creating a ‘cold’ or immune-excluded tumour, less likely to benefit from ICI. Immunosuppressive cells such as tumour-associated macrophages (TAMs), myeloid-derived suppressor cells (MDSCs), CD4 positive regulatory T cells (Tregs), and stromal cells through the combinatorial action of cell surface and secreted molecules such as immunoregulatory cytokines, chemokines, and prostaglandins can block T-cell infiltration and activation and can directly inhibit cytotoxic T-cells from attacking cancer cells. Each cell in the tumour microenvironment including tumour cells can be considered immunomodulatory typically expressing and secreting a wide variety of pro- or anti-inflammatory signalling molecules. Cytokines and chemokines are small, secreted proteins that immune cells use to communicate with each other through specific and cell type selective ligand-receptor interactions creating a complex and dynamic cell-cell interaction network. Cytokines can have activating and inhibitory effects on immune cells and non-immune cells in the TME and can through their collective effects determine the overall chemokine contexture of the TME affecting the recruitment and migration of either immunosuppressive immune cell populations or anti-tumour effector T-cell populations [7,8]. Chemokines which are a subset of cytokines which direct chemotaxis or the migration of (immune) cells in and out of tissues are emerging as key determinants of the overall immune contexture of the TME and the likelihood of a positive response to ICI [9,10]. Circulating cytokines and chemokines can be measured in serum or plasma and may be predictive of response to ICI [11,12]. The same signalling molecule may have a multitude of functions depending on context, i.e., pro-inflammatory cytokines such as tumour necrosis factor-alpha (TNF-α) may lead to tumour growth initially and can also contribute to irAEs as indicated by recent studies on the role of TNF and the potentially protective effect of anti-TNFs as an approach to ameliorate irAEs and potentiate the efficacy of ICI [13,14,15,16].

The complexity of the interactions between cancer cells, stromal cells, immune cells, extracellular matrix, and microbial cells of the tumour microbiome and the spatial and temporal variation of gene and protein expression within different parts of the sampled tumour—which may change with treatment-make the development of a single pre-treatment tissue biomarker challenging and perhaps unrealistic. Pragmatically, it is easier to obtain repeated blood samples and blood may provide an averaged snapshot of the interplay between all cellular and molecular components of the TME at multiple sites [17,18]. It is possible that the most active site of metastatic disease will contribute most to detectable components in blood such as circulating tumour DNA (ctDNA) [19].

## 2. Approved Predictive Biomarkers of Response

### 2.1. Immunohistochemistry for PD-L1

There are several approved immunohistochemistry (IHC) antibodies that detect the percentage of tumour cells expressing PD-L1. These have some predictive value of response especially in non-small cell lung cancer (NSCLC). Limitations include interobserver variation which can be improved by training and limiting PD-L1 reporting to pathologists who are experts in cancers of the organ in question, i.e., thoracic pathologists to score lung tissue and genitourinary pathologists to score bladder [20,21]. It is still essential for the pathologist to decide what the tissue of origin and the cell type of the cancer is. In lung cancer, the choice of cytotoxic chemotherapy to partner immunotherapy relies on accurate histological diagnosis [22].

The tumour proportion score (TPS) is the number of tumour cells with positive membranous staining for PD-L1 divided by the total number of viable tumour cells multiplied by 100%. The TPS cut-off value of ≥50% was derived using training and validation sets in a large single-arm trial of patients with metastatic NSCLC treated with Pembrolizumab monotherapy [23]. The area under the ROC curve was 0.743, sensitivity 70.4%, and specificity 79%. Its use is rendered more complicated by different drug companies and trials using different proprietary companion diagnostic monoclonal antibodies to PD-L1 which are not interchangeable [24]. In general, first-line Pembrolizumab monotherapy is considered in NSCLC with a PD-L1 TPS score of 50% or more (at least 50% of tumour cells express PD-L1), whereas a score of 1–49% leads to combination ICI and cytotoxic chemotherapy in non-driver mutation NSCLC [25,26] (Figure 2).

Patients with adverse prognostic features and a high TPS may be considered for first-line cytotoxic and ICI combination, especially if fit for conventional chemotherapy [27,28]. The net effect is that most patients with Stage IV NSCLC will receive first-line Pembrolizumab whether alone or in combination. PD-L1 TPS is useful when deciding on first-line monotherapy vs. combination therapy with cytotoxics in NSCLC but of limited value in second-line post exposure to chemotherapy. In general, the European Medicines Agency requires higher proof of benefit than the FDA [29] (Table 1).

The combined positivity score (CPS, also known as composite proportion score) incorporates PD-L1 expression on tumour infiltrating immune cells as well as cancer cells and has been shown to be of value in predicting response in cancers of the stomach and oesophagus [37]. CPS = # PD-L1 staining cells (tumour cells, lymphocytes, macrophages)/total # of viable tumour cells × 100. CPS is useful when considering whether to use second line immunotherapy or chemotherapy in pre-treated upper GI malignancies (Table 2) but of limited clinical value in deciding which first line patients should receive combination therapies. Pembrolizumab in combination with chemotherapy is licenced in triple negative breast cancer with a CPS score of 10 or higher [38].

PD-L1 score can vary depending on whether the primary tumour or metastasis is sampled and whether the edge of the tumour or centre is used. Core biopsies taken from lung cancer resection specimens show significant intra-tumoural heterogeneity both of tumour and immune cells [46,47]. PD-L1 staining may also vary over time and with treatment [48,49,50,51]. PD-L1 staining is relatively cheap (100–200 euro per specimen) and readily available in most cancer centres. The future of PD-L1 IHC as a biomarker lies in its incorporation into multi-parametric models. Artificial Intelligence may be used to interpret digitized images of tumour tissue including PD-L1 staining which may yield more accurate TPS and CPS [52].

### 2.2. Tumour Mutational Burden

Tumour mutational burden (TMB) is a measure of how many point mutations per one million bases or Megabase (Mb) of DNA are found in the tumour genome. It is expressed as the total number of somatic or acquired mutations per coding area of a tumour genome in mutations per Megabase (Mut/Mb). The more mutations the more neo-antigens are potentially coded for and presented to the immune system by the tumour cell via Major Histocompatibility Complex I (MHC I, also known as Human Leukocyte Antigen I (HLA I)). TMB can be calculated using whole exome sequencing (WES) or by looking at the mutational frequency in a smaller panel of affected genes using next generation sequencing (NGS). TMB correlates with response, is independent of PD-L1 expression, and has been approved by the USFDA as a tissue agnostic biomarker for the PD-L1 inhibitor pembrolizumab (TMB > 10 mt/Mb). Tumours with high TMB (>20 mt/Mb) have an approximately 45% response rate to immunotherapy [53]. TMB based on analysis of tumour tissue is not routinely available outside of clinical trials although is offered as part of a NGS package by several companies. When three commercially available platforms were used to evaluate 96 samples there was good concordance, especially in PD-L1 low samples, although cut-offs had to be altered to increase sensitivity when compared with the gold standard assay [54]. It has been shown to predict a population of NSCLC patients who respond to dual immunotherapy rather than conventional first line chemotherapy [55]. Tissue TMB suffers from the same issues of spatial and temporal heterogeneity as PD-L1 IHC [56,57]. It is hoped that blood TMB of cell-free tumour DNA (ctDNA) may provide a better biomarker [22,58]. Blood TMB has been shown to differentiate between responders and non-responders to first line Atezolizumab in NSCLC [59,60].

A complete review of the subject demonstrates that nonsynonymous mutations which lead to frameshifts and accumulation of multiple abnormal amino acid sequences when transcribed and translated are much more immunogenic than synonymous point mutations [61]. Standard TMB does not differentiate between these different types of mutations [62]. Clonal TMB, i.e., nonsynonymous mutations found in at least 95% of cancer cells is a better predictor of ICI response than sub-clonal TMB. Clonal TMB can be combined with other genetic biomarkers of favourable response such as low genetic heterogeneity, dinucleotide variants, loss of TNF signalling gene (TRAF2) [63,64]. The relatively high cost of TMB, complicated laboratory and computational analysis required, and its lack of predictive value have limited its widespread clinical adaptation outside the USA. A recent review of the subject highlights the variable success of high TMB in predicting response with huge variation between different tumour types [65]. High TMB is of value in lung cancer, where a subset of patients with absent PD-L1 expression will respond to ICI, and endometrial cancer (47% response rate), but of no discriminatory value in anal cancer. In glioma, high TMB may predict resistance to ICI. Some cancers respond well to ICI despite having low TMB such as Merkel cell carcinoma and renal cell carcinoma.

### 2.3. Microsatellite Instability

Microsatellites are short sequences of base pair repeats which may become elongated or duplicated during DNA replication in the absence of the DNA mismatch repair (MMR) complex which repairs these errors. Tumours with loss of MMR genes, whether inherited as in hereditary non-polyposis colon cancer (HNPCC or Lynch Syndrome), or sporadically acquired, exhibit genetic microsatellite instability (MSI). These MSI-high tumours are vulnerable to treatment with ICI with high response rates and durable clinical benefit [66]. Fifteen percent of sporadic colon, endometrial and gastric cancers are MSI-H. IHC to detect loss of any of the 4 MMR proteins is quick, cheap, and readily available. (Figure 3). It is reliable in colon, gastric and endometrial cancer but may be less reliable in less common MSI-H cancers such as urothelial. PCR is used to compare microsatellites in tumour and normal tissue from the same patient and is reliable across tumour types [67,68]. It may detect an extra 5–11% of MSI malignancies without loss of MMR protein expression. NGS can also be used to detect MSI and other predictive mutations but is expensive [69]. However, this is changing rapidly and NGS for MSI may become standardised and readily available [70]. As expected, there is substantial overlap between high TMB and MSI-H tumours [71]. There is also interest in using ctDNA to detect MSI-H cancers [72].

## 3. Novel Biomarkers of Response, Resistance, and Toxicity

### 3.1. Immunohistochemistry

The presence of different populations of immune cells and their spatial location can be determined using IHC. Tumours which “exclude” CD8^+^ T cells, keeping them at the edge of tumoral tissue are more likely to be resistant than those where the T cells freely infiltrate the tumour. The Immunoscore uses digital pathology to quantify CD3 and CD8^+^ T cells in the tumour and in the invasive margin of cancers. It is prognostic and predicts benefit to adjuvant cytotoxic chemotherapy. While it is best described in colorectal cancer (CRC) it has been used to predict outcomes in other malignancies [73,74,75]. It is unknown whether Immunoscore can predict benefit of ICI therapy. The Phase II POCHI trial uses Immunoscore to select patients with metastatic CRC microsatellite stable tumours and a high immune infiltrate for treatment with chemotherapy and ICI [76].

PD-L2 is another ligand of PD-1 expressed by tumour cells to escape immune attack and is under investigation as a biomarker of prognosis post-surgery [77,78]. PD-L2 is also expressed by a variety of stromal and APCs. IHC has been used to examine tumour PD-L2 staining and may predict responses to Pembrolizumab [79]. It has been suggested that ICI targeting PD1 may be more efficacious than those targeting PD-L1 in part because they interrupt PD-L2 binding [80]. A recent review article fully discusses the role of PD-L2 as a predictive biomarker and as a novel therapeutic target [81].

There are multiple other inhibitory and stimulatory molecules in the tumour cell immune synapse which are under investigation [82]. Lymphocyte-associated gene 3 (LAG3), a CD4 homologue, is another T cell surface checkpoint molecule implicated in T cell exhaustion and a therapeutic target under investigation, usually combined with anti-PDL1 treatments [83,84]. LAG3 binds to MHCII on APC and downregulates T cell function, but it has several other ligands including galectin-3, liver sinusoidal cell lectin (expressed by melanoma cells) and Fibrinogen-like Protein I [85,86]. It is unknown whether IHC for LAG3 on tumour infiltrating lymphocytes is an effective biomarker for ICI response, but it may reflect immune surveillance and better prognosis in early-stage disease [87]. Patients with PD-/LAG3 expressing melanoma whose disease had become refractory to anti PD-1/PD-L1 treatment responded better to combination of Nivolumab with Relatimab, a LAG3 targeting monoclonal antibody. This combination has been shown to be highly efficacious in first line metastatic melanoma with PFS more than doubling from 4.6 to 10.1 months, albeit at the cost of increased irAE from 9.7% to 18.9% [88]. Patients whose tumours were LAG3 low (<1%) by IHC did not benefit from addition of Relatimab [89].

Tumour cells resistant to PD-1 blockade produce IFNβ and ATRA (all-trans retinoic acid), increasing CD38 expression on the tumour cell surface. This in turn leads to production of adenosine which inhibits CD8^+^ T cells in the TME [90]. While monoclonal antibodies targeting CD38 are used in multiple myeloma they have not proved successful in solid tumours when combined with ICI in early phase trials. However, agents targeting the adenosinergic pathway may hold promise, especially in tumours with high CD38 expression detected by IHC.

As therapies targeting other immune checkpoints such as T cell immunoglobulin and mucin 3 (TIM3) and T cell immunoreceptor with immunoglobulin and ITIM domain (TIGIT) enter clinical practice IHC of tumour and immune tissue for these and other targets may play a role [82,91]. It is likely that these will be combined with a backbone of anti PD-1/PD-L1 ICI [92,93].

### 3.2. Systemic Markers of Inflammation

It is hypothesized that patients with a low burden of disease, whose immune system is already attempting to attack the tumour will have the best outcomes. There are several routine blood tests that can be used to estimate tumour burden (e.g., lactate dehydrogenase) and beneficial immune activation (e.g., elevated lymphocyte count) [94]. There are also markers of systemic inflammation which portend worse outcomes including elevated neutrophil and platelet counts and C-reactive protein (CRP), which reflects interleukin-6 (IL6) levels [95,96,97]. These can be combined to create systemic inflammatory scores and possibly to guide treatment decisions. One example is the systemic immune-inflammation index (SII) which is calculated as (neutrophils × platelets/lymphocytes). Elevated SII is associated with poor outcomes in patients with pancreatic cancer treated with ICI and other therapies [98]. Studies have shown that male obese melanoma patients did better than non-obese patients with ICI [99]. Obesity (BMI over 30) drives tumorigenesis and PD-L1 expression but makes these tumours vulnerable to ICI without an increase in irAE [100].

### 3.3. Cytokines, Chemokines, and Other Soluble Immune Markers

Different cytokine and chemokine cell-cell interaction networks in the TME, tumour draining lymph nodes and tertiary lymphoid structures will dictate the overall spatial composition of the immune cell component of the TME, its contribution to the overall tumour Immunoscore and the classification of the tumour as ‘hot’, ‘cold’ or ‘immune excluded’. This in turn may predict different therapeutic outcomes depending on the type of tumour and therapy used. Many studies are contradictory and large data sets will be needed to elucidate how best to use baseline tumour and systemic cytokine levels, gene expression signatures, or on treatment changes as indicated by this renal cancer study [101]. Tumour necrosis factor-alpha (TNF-α) and Interferon-gamma (IFN-γ) released by CD4^+^ Th1 T cells and CD8^+^ cytotoxic T cells trigger senescence and cell death in tumour cells and efficacy of ICI is dependent on tumour cell responsiveness to IFN-γ as evidenced by the emergence of adaptive resistance to ICI in some patients mediated through accumulated mutations in genes coding for JAK-STAT signalling and antigen processing and presentation [102,103,104,105]. Transforming growth factor beta (TGF-β) is an immunomodulator thought to restrain the immune system. It would be logical to assume that high baseline circulating TGF-β would be linked with poor responses, but it may be a signal that the immune system is actively trying to attack the tumour, which is protected by TGF-β as inflammatory and immunoregulatory responses are closely coordinated and can be tipped towards success by ICI. IL-17 is known to be associated with autoimmune colitis [106]. Pre-treatment levels of cytokines in melanoma patients treated with neoadjuvant ipilimumab (CTLA4 inhibitor) have different risks of toxicity (IL-17 high) and resistance (TGF-beta1 low and IL-10 high). In metastatic melanoma responders to nivolumab had elevated baseline TGF-β [107,108]. Other studies have found high baseline TGF-β predicts poor outcomes in hepatocellular carcinoma treated with Pembrolizumab [109]. Circulating cytokines and chemokines (CXCL2 and CXCL5) have been shown to predict response and immune toxicity in patients treated with anti-PD-L1 therapy [110,111]. CXCR2 and IL2ra increased from baseline in a patient treated with nivolumab who developed radiation pneumonitis and levels of these cytokines declined on initiation of corticosteroids [112]. While changes in cytokine levels can be observed in patients treated with Atezolizumab they do not appear to segregate responders from non-responders [113]. Splice variants of PD-L1 are shed from cell surfaces and can be measured in blood samples [114]. High soluble PD-L1 (sPD-L1) is associated with worse outcomes in lung but not gastric cancer [115,116]. A meta-analysis of multiple non immunotherapy trials in multiple tumour types found sPD-L1-high to be associated with worse overall survival (OS) [117]. Soluble PD-L1 has been reported to have diametrically opposed in vitro functions by different researchers [118]. Other immune checkpoint transmembrane receptors can be cleaved by metalloproteinases and shed into the circulation such as CTLA-4, LAG-3, and TIM-3. In the setting of ICI therapy low baseline sPD-L1 is associated with better outcomes and high or increased levels at two months with treatment failure [119,120]. Early clinical data suggest that high levels of circulating cleaved LAG3 (sLAG3) may predict immunotherapy response [121].

### 3.4. Immune Metabolism

Activated T cells are highly metabolically active and consume tryptophan to generate kynurenine which is an immunosuppressant [122]. Increases in the kynurenine/tryptophan (K/T) ratio at 4–6 weeks but not the baseline K/T ratio have been shown to predict resistance to ICI in renal cell carcinoma (RCC) and melanoma [123]. Melanoma patients treated with Nivolumab whose K/T ratio increased >50% had a median OS of 15.7 months compared to >38 months in those with falling K/T ratio. Rises in K/T ratio correlated with increased PD-L1 expression in RCC patients and worse OS receiving Nivolumab but not Everolimus suggesting a potential immunotherapy resistance mechanism. Li et al. suggest that while a high baseline K/T ratio is a prognostic marker of reduced survival reflecting disease bulk that dynamic changes in K/T ratio reflect a predictive biomarker of ICI response. The enzyme indoleamine-2,3 dioxygenase (IDO1) is the first step of tryptophan catabolism. It was hoped that drugs such as Epacadostat which inhibit IDO1 would be synergistic with ICI but so far this approach has failed in clinical trials [124,125]. Possibly these drugs should be reserved for patients with rising K/T ratio post ICI exposure. Metformin may have synergistic effects when combined with ICI [126]. Whether or not manipulating immune and tumour metabolism is a useful therapeutic target changes in the K/T ratio may still be a useful biomarker.

### 3.5. Flow Cytometry of Circulating Immune Cells

Immune cells constantly move in and out of the tumour microenvironment and circulate in peripheral blood and lymph nodes. It is challenging to perform flow cytometry of tissue specimens which require rapid processing of fresh or frozen samples and the perennial sampling issues of all tissue biomarkers apply. Peripheral blood may hold greater utility, especially when comparing baseline to early treatment samples. Flow cytometry has been used to identify different populations of immune cells both in the TME and in peripheral blood [127,128]. In general, MDSC protect the tumour from immune attack, as do Tregs. CD4^+^ T regs also express surface CD25 and the transcription factor FOXP3. Other T helper cells (Th1 subtype (secretes IFN-γ and IL-2)), and IL-9 and IL-10 producing T cells (Th9 cells) have anticancer properties [129]. A rise in Th9 circulating cells is associated with melanoma response to nivolumab [108]. The complexity of analysing peripheral blood mononuclear cells (PMBCs) can be seen in the case of CD14^+^ monocytes. Pre-treatment levels of HLA-DR expression on the monocyte surface appear to have a predictive effect with low levels indicating an immunosuppressive phenotype and worse outcomes whereas HLA-DRhi cells correlate with improved outcomes [130,131]. Huang et al. described how early changes in circulating T cell populations were a powerful predictor of response in melanoma patients, especially combined with imaging estimates of tumour burden [132]. In lung cancer, circulating lymphocytes with high-level PD-1, PD-L1, and PD-L2 expression are associated with poorer prognosis when treated with cytotoxics and tyrosine kinase inhibitors [133]. CD8^+^ T cells attempt to bind to and destroy tumour cells but are inhibited by PD-1/PDL-1 signalling. These frustrated CD8^+^ T cells become dormant or exhausted T cells, expressing PD-1, TIM, ICOS and LAG3. Prior work has mostly investigated the pre-treatment ratio of effector (CD8^+^) to inhibitory cells (Tregs and MDSCs) as a predictor of outcome. Some studies have shown that early increases in circulating effector T cells (Ki67^+^, PD-1^+^, CD8^+^) post ICI are predictive of lung cancer response [134]. Other work has examined reactivation of exhausted T cells measured by Ki67 expression. Autoimmune toxicity of combined (anti-CTLA4 and PD-1) ICI is associated with early activation (Ki67) of CD8^+^ effector and memory cells [135]. CD27 and CD28 are expressed by activated T cells and expression normally decreases as cells mature. Low pre-treatment levels of these surface markers at baseline are associated with decreased risk of irAE [135].

### 3.6. Next Generation Sequencing

ct DNA can be used to quantitate tumour burden and its early fall at 3–6 weeks is an on-treatment predictor of lung cancer ICI response [136]. Patients whose ctDNA remains detectable at 12 weeks have a worse prognosis [137]. In patients who are responding to treatment, rises in ctDNA may precede clinical progression and potentially identify resistance mechanisms. The maximum somatic allele frequency (MSAF), or proportion of total circulating DNA derived from cancer, combined with bTMB retrospectively predicted response to atezolizumab versus docetaxel in two large trials of atezolizumab in NSCLC. In particular, a group with worse outcomes when treated with immunotherapy (bTMB low and MSAF-high) could be defined [138]. Copy number loss is associated with ICI resistance probably caused by decreased expression of genes involved in antigen expression and tumour cell IFN-γ signalling [135,139]. Tissue NGS can identify tumour genes predictive of response and resistance (mutations in STK11 and KEAP1 in RAS-mutated NSCLC, loss of MHC Class I expression, impaired IFN receptor signalling) [140,141]. One small study found that p53 mutated NSCLC was more responsive to ICI [142]. Gene expression profiling (GEP) of tumour tissue, looking for differing levels of mRNA, holds great promise. Early changes in tissue gene expression have been shown to predict response in mouse models [143]. It may be possible to use GEP of TME across different solid tumour types [144]. GEP of tumour cells has been shown to predict benefit of ICI in small cell lung cancer (SCLC) [145].

All components of the TME (cancer, immune and stromal cells) can be sampled. EGFR mutated NSCLC is resistant to ICI however a recent study indicated that patients whose tissue GEP indicated an inflamed TME may derive some benefit [146]. In the CheckMate 275 study of Nivolumab in urothelial cancer high expression of genes associated with epithelial to mesenchymal (EMT) transformation predicted an adverse outcome whereas high expression of genes associated with Type I Interferon immune response predicted favourable outcomes [147,148]. The promise of mRNA is not confined to tissue and GEP of circulating tumour and immune RNA can also be used. Acquired resistance to ICI can be mediated by mutations in cancer cells that are selected under immune pressure. These mutations include loss of interferon receptor downstream signalling by loss of function mutations in JAK1 and JAK2. Beta 2 microglobulin (B2M) is essential for MCH I function and presentation of antigen to T cells by tumour cells [149]. Loss of B2M is associated with immunotherapy resistance. PTEN loss predicts ICI resistance [150]. Cyclin D1 (CCND1) amplification down-regulates PD-L1 expression and is associated with reduced ICI response [63]. T cell receptor diversity or repertoire (the variation in the binding region of different T cell clones) can be measured using NGS in tumour tissue and in blood [151]. Increased diversification of the T cell repertoire at 2 weeks is associated with irAE in patients treated with anti-CTLA-4 [152]. Clonal expansion of T cells measured by sequencing TCR beta chains also precedes Ipilimumab toxicity [153]. Drugs which enhance thymic function and TCR diversity may play a role in improving response rates to ICI [154].

### 3.7. Microbiome as Biomarker and Therapeutic Target

We live in harmony with trillions of bacteria in our gut, which influence our immune system in a dynamic fashion (Figure 4). Circulating CD8^+^ T cells are primed by CD68^+^ APC in the gut lymph nodes, which in turn are influenced by interactions with gut bacteria and bacterial metabolites. Gut microbiota release metabolites with immunomodulatory activities such as vitamin B and short-chain fatty acids (SCFAs). Several studies have shown that patients who have not received antibiotics one to three months prior to treatment and who have a healthy diverse faecal microbiome have a better response to ICI in melanoma and lung cancer [155,156,157].

Antibiotics prior to ICI therapy appear to reduce the incidence of ICI-mediated colitis but may make colitis more severe if initiated post ICI [158]. Immune-mediated colitis resulting from ICI treatment has successfully been treated with faecal microbiota transplant (FMT) in humans [159]. Tumour samples from cervical cancer patients with a more diverse microbiome are more heavily infiltrated by T lymphocytes [160]. In mouse models, faecal human microbiota transplant (HMT) from human responders to germ-free mice caused tumour shrinkage compared to HMT from non-responders [161]. In mouse models of cancer, intestinal *Bifidobacterium pseudolongum* produced inosine, which could leak into the circulation and activate cytotoxic T cells via adenosine receptors when ICI caused decreased gut barrier function [162]. Certain bacterial genera such as *Akkermansia*, *Faecalibacterium*, *Ruminococcaceae*, and *Bacteroides* are associated with ICI response, whereas *Bacteroidales thetaoitaomicron* and *Escherichia coli* are associated with resistance [163,164]. Patients treated with dual ICI with a higher abundance of *Bacteroides intestinalis* had more severe irAE. In those patients with colitis, and in mouse models, IL-1β was upregulated in colonic biopsies [135]. Different groups have reported varying positive and negative associations with individual species but there appear to be clear differences in the faecal microbiota of responders and non-responders [165]. It is unlikely that a single microbiome score will be prognostic or predictive, but it may well be incorporated into multiparametric models. FMT clinical trials in humans were placed on hold after adverse events but have recently been shown provide benefit to some non-responders [166,167]. Probiotics and faecal viral transfer may play a role in therapeutic modulation of the microbiome, but this is under investigation and should not be attempted outside clinical trials [168,169,170]. Geographic location, genetics, ethnicity, diet, age and medication and environment all influence the microbiome and will have to be taken into account if the faecal microbiome is to be used as a biomarker or manipulated therapeutically [171,172]. The microbiome is usually characterised using 16S ribosomal RNA sequencing, but more information can be obtained using NGS sequencing, which is becoming standard in microbiome research [173]. The enormous amount of data generated by metagenomics looking at which bacterial metabolic pathways are activated, can be analysed using machine learning tools such as random forest analysis [174].

## 4. Conclusions

Previous attempts to escalate ICI from mono to dual-therapy based on imaging changes were unsuccessful. Multi-parametric models which combine some or all of the above may enable us to choose which patients are treated with anti-PD-1 or PD-L1 monotherapy (low total ctDNA, high bTMB, favourable immune profile) or combined anti-CTLA4 and anti-PD-1 or PD-L1 (high ctDNA, low bTMB, unfavourable immune profile). Immune-PET may also play a role [175]. Patients who are at increased risk of toxicity may be treated with prophylactic targeted immune suppressants such as anti-TNF monoclonal antibodies which have been shown to paradoxically enhance ICI response. ICI have deepened our knowledge of the immune system in health and disease [14,15]. It is possible that combinations of antiPD-1/PD-L1 agents with antibodies to novel checkpoints, such as LAG3, will become a standard of care where affordable. As novel agents targeting other checkpoints such as TIGIT and LAG3 make their way out of trials into practice our patients deserve accurate biomarkers to guide their use [88,91,176]. Both small sample biomarker discovery trials and translational studies associated with large commercial and academic clinical trials have given us a glimpse of how we may in the future be able to select treatments according to the patient’s individual immune system, including early modification of treatment based on rise or fall of cytokines, circulating immune cell populations and ctDNA. To make this a reality, scientists and clinicians should standardize how they measure and report clinical data wherever possible [18,177].

## Figures and Tables

**Figure 1 diagnostics-12-00124-f001:**
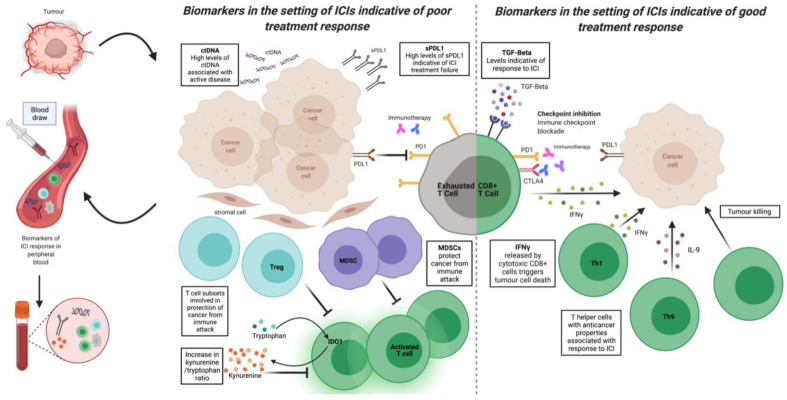
Tumour microenvironment and novel potential biomarkers: sPDL1 = soluble PDL1; ctDNA = circulating tumour DNA; TGF-B = transforming growth factor beta; Trp = tryptophan; IDO = indoleamine 2,3-dioxygenase; Treg = regulatory T cell; Tex = exhausted T cell; MDSC = myeloid derived suppressor cell; Th1 = T helper cell Th1 subtype; Th9 = T helper cell Th9 subtype; IFNy = interferon gamma; IL-9 = interleukin 9; ICI = immune checkpoint inhibitor.

**Figure 2 diagnostics-12-00124-f002:**
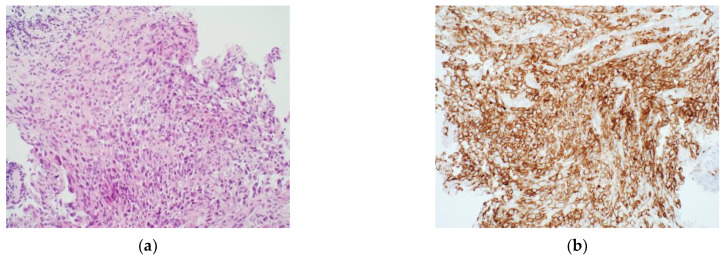
Example of H&E and PD-L1 IHC in NSCLC: (**a**) 20× non-small cell lung cancer stained with haematoxylin and eosin. (**b**) Same specimen stained for PD-L1. Tumour Proportion Score = 100%. Provided by R.F., S.J.H.

**Figure 3 diagnostics-12-00124-f003:**
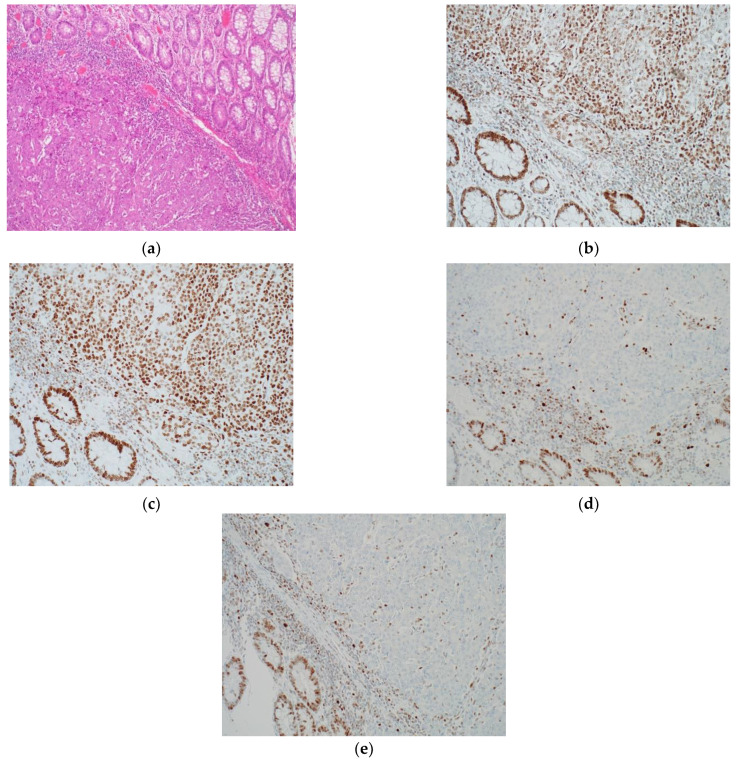
Examples of IHC as an Immunotherapy Biomarker in Colorectal Cancer: (**a**) Colorectal cancer H&E 10× magnification; (**b**) CRC intact expression of MSH2 20×; (**c**) MSH6 expression 30×; (**d**) loss of MLH1 20×; (**e**) loss of PMS2 20×. Micrographs provided by R.F. S.J.H.

**Figure 4 diagnostics-12-00124-f004:**
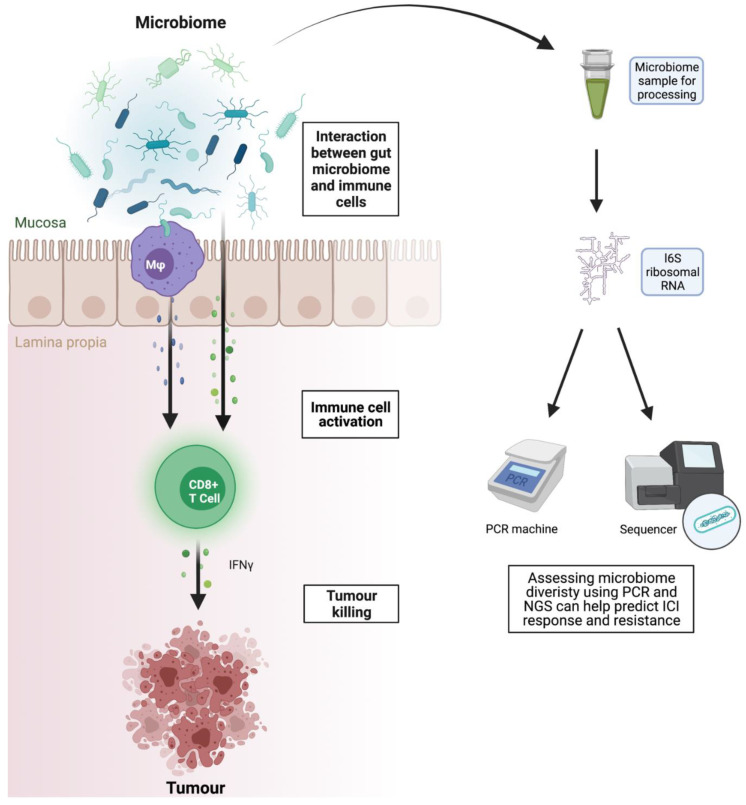
Assessing faecal microbiome diversity and composition as a biomarker of response and toxicity. Depiction of the interaction between the gut microbiome, immune cells and the tumour microenvironment. Cytokines released from activated immune cells promote tumour killing. Targeted PCR and next generation sequencing of microbiome samples can be used to assess microbiome diversity, which may predict response to immune checkpoint inhibition. Mφ = macrophage; IFNγ = interferon gamma; PCR = polymerase chain reaction; NGS = next generation sequencing.

**Table 1 diagnostics-12-00124-t001:** Tumour proportion score (TPS) using Dako 22C3 immunohistochemistry as a biomarker of Pembrolizumab efficacy in first-line metastatic EGFR and ALK wild type NSCLC compared to cytotoxic chemotherapy alone. Trials were selected to show the evolution of pembrolizumab from salvage to first line therapy and the limitations of TPS.

TRIAL ARM	PD-L1 TPS (Number of Patients Receiving Pembrolizumab)	Overall Response Rate (95% CI) (%)	Median PFS in Months (95% CI)	PFS HR Compared to Chemo Alone (95% CI, *p*-Value)	Median OS in Months (95% CI)	OS HR Compared to Chemo Alone (95% CI, *p*-Value)	Approval
KEYNOTE-024 Single Agent Pembrolizumab (Pre-treated AC and SCC [25,30,31,32]	≥50% (154)	44.8	10.3	0.5 (0.37–0.68, <0.001)	26.3 (95% CI 18.3–40.4)	0.62 (0.48–0.81, 0.002)	Ireland, EMA [33], FDA
KEYNOTE-042 Single Agent Pembrolizumab (First Line AC and SCC) [26]	All Patients (636)				16.7	0.81 (0.71–0.93, 0.0036)	FDA
KEYNOTE 042 [26]	≥50% (298)	39 (34–45)	7.1 (5.9–9)	0.81 (0.67–0.99, 0.017)	20 (15–24.9)	0.69 (0.56–0.85, 0.003)	Ireland, EMA, FDA
KEYNOTE 042 [26]	≥20% (412)	33 (29–38)	6.2 (5.1–7.8)	0.94 (0.8–1.11,)	17.7 (15.3–22.1)	0.77 (0.64–0.92, 0.002)	FDA
KEYNOTE 042 [26]	≥1% (636)	27 (24 -31)	5.4 (4.3–6.2)	1.07 (0.94–1.21)	16.7 (13.9–19.7)	0.81 (0.71–0.93, 0.0018)	FDA
KEYNOTE 042 [26]	1–49% (338)				13.4 (10.7–18.2)	0.92 (0.77–1.11,)	FDA
Pembrolizumab and Chemotherapy (First Line AC) KEYNOTE 189 [34,35]	All Patients (410)	48 (43.1–53)	9 (8.1–9.9)	0.48 (0.4–0.58)	22 (19.5–25.2)	0.56 (0.45–0.70)	Ireland, EMA, FDA
KEYNOTE 189 [34,35]	≥50% (132)	62.1 (53.3–70.4)	11.1 (9.1–14.4)	0.36 (0.26 -0.51)	NR (20.4–NR)	0.59 (0.39–0.86)	
KEYNOTE 189 [34,35]	1–49% (128)	49.2 (40.3–58.2)	9.2 (7.8–13.1)	0.51 (0.36–0.73)	21.8 (17.7–25.9)	0.62 (0.42–0.92)	
KEYNOTE 189 [34,35]	<1% (127)	32.3 (24.3–41.2)	6.2 (4.9–8.1)	0.64 (0.47–0.89)	17.2 (13.8–22.8)	0.52 (0.36–0.74)	
Pembrolizumab and Chemotherapy (First Line SCC) KEYNOTE-407 [36]	All Patients (278)	57.9% (51.9–63.8)	6.4 (6.2–8.3)	0.56 (0.45–0.70; <0.001)	15.9 (13.2–NE)	0.64; (0.49–0.85; <0.001)	Ireland, EMA, FDA
	≥50% (73)	60.3 (48.1–71.5)	8.0 (6.1–10.3)	0.37 (0.24–0.58)	NR (11.3–NE)	0.64 (0.37–1.10)	
	1–49% (103)	49.5 (39.5–59.5)	7.2 (6.0–11.4)	0.56 (0.39–0.8)	14.0 (12.8–NE)	0.57 (0.36–0.90)	
	≥1% (183)			0.49 (0.38–0.65)			
	<1% (95)	63.2 (52.6–72.8)	6.3 (6.1–6.5)	0.68 (0.47–0.98)	15.9 (13.1–NE)	0.61 (0.38–0.98)	

Table 1 Legend; immunohistochemistry (IHC) progression-free survival (PFS), hazard ratio (HR), overall survival (OS), confidence interval (CI), tumour proportion score (TPS), *p*-value only listed if significant, not reached (NR), not estimable (NE). Most recent publication used where applicable. Note that the populations in KEYNOTE 042 overlap and that most of the benefit is in the group with TPS ≥ 50%. European Medicines Agency, Food and Drug Administration (USA). Ireland—funded by National Cancer Control Program in this indication. Not estimable (NE). Adenocarcinoma (AC), squamous cell carcinoma (SCC).

**Table 2 diagnostics-12-00124-t002:** Trials showing the use of the combined positivity score (CPS) in HER2 negative upper gastrointestinal malignancies.

TRIAL ARM	PD-L1 CPS (No. of Patients Receiving Pembrolizumab or Nivolumab)	Overall Response Rate (95% CI) (%)	Median PFS in Months (95% CI)	PFS HR Compared to Chemo Alone (95% CI, *p*-Value)	Median OS in Months (95% CI)	OS HR Compared to Chemo Alone (95% CI, *p*-Value)	Approval
KEYNOTE-061 (pre-treated gastric and GOJ cancer Taxol vs. Pembro) [39]	All patients (296)	11	1.6	1.34 (1.12–1.60)	6.7 (5.4–8.9)	0.94 (0.79–1.12)	
KEYNOTE-061	CPS ≥ 1 (196)	16	1.5	1.27 1.03–1.57)	9.1	0.82, 0.66–1.03; one-sided *p* = 0.0421	FDA (gastric cancer 3rd line)
KEYNOTE-061	CPS ≤ 1 (99)	2					
KEYNOTE-061 (post-hoc analysis)	CPS ≥ 10 (53)	24.5					FDA (gastric cancer of GOJ cancer 2nd line)
KEYNOTE-062 (Pembro alone vs. Pembro + Chemo vs. Chemo alone first line gastric AC) [40]	Pembro Alone (256)	14.8			10.6 (7.7–13.8)	0.91 (99.2% CI 0.69–1.18)	
	Pembro Alone CPS > 1		2 (1.5–2.8)	1.66 (1.37–2.01)		0.91 (0.74–1.1)	
	Pembro Alone CPS ≥ 10	23	2.9 (1.6–5.4)	1.10 (0.79–1.51)	17.4 (9.1–23.1)	0.69 (0.49–0.97)	
	Pembro + Chemo (250)	37.2			12.5 (10.8–13.9)	0.85 (0.7–1.03)	
	Pembro + Chemo CPS ≥ 1						
	Pembro + Chemo CPS ≥ 10		6.9 (5.7–7.3)	0.84 (0.7–1.02)	12.3 (9.5–14.8)	0.85 (0.62–1.17, 0.16)	
KEYNOTE-180 (Pre-treated Oesophageal AC and SCC) [41]		9.9 (5.2–16.7)					
	CPS < 10 (63)	6 (2–16)	2.0 (1.9–2.1)		5.4 (3.9–6.3)		
	CPS ≥ 10 (58)	14 (6–25)	2.0 (1.9–2.2)		6.3 (4.4–9.3)		
	SCC CPS ≥ 10 (35)	20					FDA
KEYNOTE-181 (pre-treated AC and SCC Pembro vs. chemo) [37]	All Patients (314)	13.1 (9.5–17.3)	2.1 (2.1–2.2)	1.11 (0.94–1.31)	7.1	0.89 (0.75–1.05, 0.0560)	
	CPS ≥ 10	21.5 (14.1–30.5)	2.6 (2.1–4.1)	0.73 (0.54 to 0.97)	9.3 (6.6–12.5)	0.69 (0.52–0.93, 0.0074)	FDA
	SCC	16.7 (11.8–22.6)	2.2 (2.1–3.2)	0.92 (0.75–1.13)	8.2	0.78 (0.63–0.96, 0.0095)	
	SCC CPS < 10	11.9	2.1 (2.1–2.4)		7.3 (5.7–9.2)		
	AC CPS < 10	3.3		2.1 (1.9–2.1)	5.1 (4.1–7.1)		
ATTRACTION-2 (pretreated gastric or GOJ AC, Nivo vs. Placebo) [42]	N/A (493 received Nivo)	11.9	1.61 (1.54–2.30)	0.60 (0.49–0.75, < 0.0001)	5.26 (4.60–6.37)	0.62 (0.51–0.76, *p* < 0.0001)	FDA
ATTRACTION-3 (pre-treated oesophageal SCC, Nivo vs. placebo) [43]	N/A (210 received Nivo)				10.9 (9.2–13.3)	0.77 (0.62–0.96, 0.019)	FDA, EMA
KEYNOTE -590 (Advanced first line oesophageal or GOJ cancer, chemo and Pembro, 73.5% SCC, 25.5% AC) [44]	All patients (373)	45.0 (39.9–50.2)	6.3 (6.2–6.9)	0.65 (0.55–0.76; *p* < 0.0001).	12.4 (10.5, 14.0)	0.73 (0.62–0.86, <0.0001)	FDA
	SCC CPS ≥ 10 (143)		7.3 (6.2–8.2)	0.53 (0.40–0.69)	13.9 (11.1–17.7)	0.57 (0.43–0.75, <0.0001)	FDA,
	All SCC (274)		7.5		12.6 (10.2–14.3)	0.72 (0.60–0.88, 0.0006)	FDA
	All CPS ≥ 10 (186)		7.5 (6.2–8.2)		13.5 (11.1–15.6)	0·62 (0.49–0.78, *p* < 0.0001)	FDA,
	AC CPS ≥ 10 (43)		8.0 (6.0–8.3)	0.49 (0.30–0.81)	12.1 (9.6–18.7)	0.83 (0.52–1.34)	FDA, EMA
	AC CPS < 10 (54)		6.3 (5.6–8.3)	0.76 (0.49–1.19)	12.7 (8.1–16.1)	0.66 (0.42–1.04)	
CheckMate 649 (Nivo + chemo vs. chemo alone in first line gastric, GOJ, oesophageal AC) [45]	All patients (603)	58 (54–62)	7.7 (7.1–8.5)	0.77 (0.68–0.87)	13.8 (12.6–14.6)	0.8 (0.68–0.94, <0.0002)	FDA
	CPS ≥ 5 (473)	60 (55–65)	7.7 (7.0–9.1)	0·68 (98% CI 0·56–0·81, <0·0001)	14.4 (13.1–16.2)	0.71 (98.4% CI 0·59–0·86, <0.0001)	FDA, EMA
	CPS ≥ 1 (641)	60	7.5 (7.0–8.4)	0.74 (0.65–0.85)	14.1 (11.6–15)	0.77 (99.3% CI 0.064–0.92, <0.0001)	FDA
	CPS < 5	55	7.5	0.93 (0.76–1.12)	12.4	0.94 (0.78–1.13)	FDA
	CPS < 1 (140)	51	8.7	0.93 (0.69–1.26)	13.8	0.79 (0.7–0.89)	FDA

## Data Availability

Not applicable.

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
