# Peer review of "Cancer Immunotherapy with Immune Checkpoint Inhibitors-Biomarkers of Response and Toxicity; Current Limitations and Future Promise"

_diagnostics, 2022, doi:10.3390/diagnostics12010124_

Round 1
Reviewer 1 Report
I have read this review with great interest, and I have picked up new elements. The topic is important, and the different elements taken into the content are presented in an appropriate way.
I have some minor suggestions for the authors that they can take into account.
“Immunotherapy” in the title is too broad. I know that oncologists use this popular term as well. But it is strongly misleading, as it concerns “only” modulatory immunotherapy, but not passive immunotherapy, adapted immunotherapy, or active specific immunotherapy.
At several places, exemplified in line 140, there is no difference in Legend layout versus Text layout. This should be improved in the final printable version.
Line 155-156. I propose to add a reference to the sentence “PD-L1 staining may also vary over time and with treatment”. This is very very very important. Now, CPIs are sometimes indicated because of the pathology of the original tissue, and oncologists neglect that this might change over time.
Similar in line 180-181 with the sentence “Tissue TMB suffers from the same issues of spatial and temporal heterogeneity as PD-L1 IHC”. We also need here references. It might even be placed in a broader context: if TMB increases over time and maybe due to chemotherapeutics, maybe immunotherapy becomes more important, scheduled adjuvant in addition to first line treatment.
Line 199. “Nonsynonymous point mutations”. I am not an expert in that particular domain, but it came over to me that this sentence is in contradiction to the sentence above in line 184 to 186.
Author Response
Dear Editor - I thank this reviewer for their careful reading and helpful suggestions. If you are able to put the tables into Landscape while leaving the figures and text in Portrait this may help with readability? I can upload files of the Tables saved in Landscape if this would help?
Response to Reviewer
Thank you for these thoughtful and helpful comments. I really appreciate them and they make the review stronger.
“Immunotherapy” in the title is too broad. I know that oncologists use this popular term as well. But it is strongly misleading, as it concerns “only” modulatory immunotherapy, but not passive immunotherapy, adapted immunotherapy, or active specific immunotherapy.
Response 1; I agree and have amended the title
At several places, exemplified in line 140, there is no difference in Legend layout versus Text layout. This should be improved in the final printable version.
Response 2; I changed the text of the Figure and Table Legends and inserted page and paragraph breaks. It would be great if the editors could put the tables in Landscape? If desired I can upload the tables separately as Word Landscape Files. I have attempted to insert page breaks to make the difference between the main body of text and the Figure and Table Legends clearer.
Line 155-156. I propose to add a reference to the sentence “PD-L1 staining may also vary over time and with treatment”. This is very very very important. Now, CPIs are sometimes indicated because of the pathology of the original tissue, and oncologists neglect that this might change over time.
Response 3; Agreed – I have added 4 references to provide data to back up this point.
Similar in line 180-181 with the sentence “Tissue TMB suffers from the same issues of spatial and temporal heterogeneity as PD-L1 IHC”. We also need here references. It might even be placed in a broader context: if TMB increases over time and maybe due to chemotherapeutics, maybe immunotherapy becomes more important, scheduled adjuvant in addition to first line treatment.
Response 4; Agreed- I have added 2 references to provide data to back up this point. I’m not sure that we can say that TMB increases in response to chemo yet but it is a plausible biological hypothesis.
Line 199. “Nonsynonymous point mutations”. I am not an expert in that particular domain, but it came over to me that this sentence is in contradiction to the sentence above in line 184 to 186.
Response 5; I have changed the paragraph to make it clearer that TMB as normally reported does not predict immunogenicity as well as clonal TMB which takes into account both the percentage of tumour cells with mutations and the amount of nonsynonymous point mutations. It now reads;
A complete review of the subject demonstrates that nonsynonymous mutations which lead to frameshifts and accumulation of multiple abnormal amino acid sequences when transcribed and translated are much more immunogenic than synonymous point mutations[60]. Standard TMB does not differentiate between these different types of mutations [61].
Reviewer 2 Report
Bird et al., and colleagues in the review manuscript entitled “Cancer Immunotherapy Biomarkers of Response and Toxicity; current limitations and future promise” reviewed current biomarker responses to ICI treatment such as IHC, TMB, microsatellite instability, soluble immune markers, microbiome, etc and described the advancement in assessing these biomarkers. The review is well written but can be improved as, to some extent, it lacks connectivity between sentences within each paragraph. Since the title of the review manuscript also talks about the toxicity of cancer immunotherapy, I would recommend the authors to elaboration a para or two on the toxicity/adverse events of ICI treatment and potential predictors of ICI toxicity. A column on the adverse effects from the clinical trials can also be included in the tables (Table 1 or 2) described by the authors.
Author Response
Bird et al., and colleagues in the review manuscript entitled “Cancer Immunotherapy Biomarkers of Response and Toxicity; current limitations and future promise” reviewed current biomarker responses to ICI treatment such as IHC, TMB, microsatellite instability, soluble immune markers, microbiome, etc and described the advancement in assessing these biomarkers. The review is well written but can be improved as, to some extent, it lacks connectivity between sentences within each paragraph. Since the title of the review manuscript also talks about the toxicity of cancer immunotherapy, I would recommend the authors to elaboration a para or two on the toxicity/adverse events of ICI treatment and potential predictors of ICI toxicity. A column on the adverse effects from the clinical trials can also be included in the tables (Table 1 or 2) described by the authors.
Thank you for these helpful comments.
We have deliberately placed predictors of immune toxicity towards the end of each section discussing novel biomarkers i.e. 3.3 Cytokines, chemokines, and other soluble immune markers Line 288 of the revised manuscript “Pre-treatment levels of cytokines in melanoma patients treated with neoadjuvant ipilimumab (CTLA4 inhibitor) have different risks of toxicity (IL-17 high) and resistance (TGF-beta1 low and IL-10 high).” Most of the studies which look at predictors of toxicity also look at efficacy and so we have chosen to discuss efficacy and toxicity together under the headings of different biomarkers.
The point of the tables is to show the limited utility of PD-L1 TPS and CPS as predictive biomarkers of response to ICI. The toxicities experienced by patients in each trial arm are well reported and as far as I’m aware not broken down by PD-L1 score. I think that dding toxicity data would confuse these tables
Reviewer 3 Report
This is good review on checkpoint inhibitors
PD1-PD-L1 pathway. I can recommend the following points to address:
1.Fig1,please include scale bar inside figure.
2. Please include role of PD-L2 in tumors and interaction with PD
3. Expand description of other checkpoint
pathways:LAG-3, TIGIT, TIM1, TIM-3.
Please describe several reports on combination therapy and synergy.
Author Response
This is good review on checkpoint inhibitors
PD1-PD-L1 pathway. I can recommend the following points to address:
1.Fig1,please include scale bar inside figure.
Response; I’m not sure I agree here. Most readers will have an idea of the average size of tumour and immune cells and clearly the cell surface molecules such as PD1 are not to scale. If you mean include a scale bar in the micrographs of human tissue sections I can ask my pathology colleague for one but I don’t think it’s essential.
- Please include role of PD-L2 in tumors and interaction with PD
I have expanded my discussion of PD-L2 immunohistochemistry;
PD-L2 is another ligand of PD-1 expressed by tumour cells to escape immune attack and is under investigation as a biomarker of prognosis post-surgery [77,78]. PD-L2 is also expressed by a variety of stromal and APCs. IHC has been used to examine tumour PD-L2 staining and may predict responses to Pembrolizumab [79]. It has been suggested that ICI targeting PD1 may be more efficacious than those targeting PD-L1 in part because they interrupt PD-L2 binding[80]. A recent review article fully discusses the role of PD-L2 as a predictive biomarker and as a novel therapeutic target [81].
- Expand description of other checkpoint
pathways:LAG-3, TIGIT, TIM1, TIM-3.
Please describe several reports on combination therapy and synergy.
I have expanded this in the Novel IHC section below and I have also highlighted this in the Conclusion Section
Please see the attachment
